# Taurine Enhances Iron-Related Proteins and Reduces Lipid Peroxidation in Differentiated C2C12 Myotubes

**DOI:** 10.3390/antiox9111071

**Published:** 2020-10-31

**Authors:** Ulrike Seidel, Kai Lüersen, Patricia Huebbe, Gerald Rimbach

**Affiliations:** Institute of Human Nutrition and Food Science, University of Kiel, 24118 Kiel, Germany; luersen@foodsci.uni-kiel.de (K.L.); huebbe@foodsci.uni-kiel.de (P.H.); rimbach@foodsci.uni-kiel.de (G.R.)

**Keywords:** skeletal muscle, labile iron pool, myoglobin, glutathione, BODIPY

## Abstract

Taurine is a nonproteinogenic amino sulfonic acid in mammals. Interestingly, skeletal muscle is unable to synthesize taurine endogenously, and the processing of muscular taurine changes throughout ageing and under specific pathophysiological conditions, such as muscular dystrophy. Ageing and disease are also associated with altered iron metabolism, especially when there is an excess of labile iron. The present study addresses the question of whether taurine connects cytoprotective effects and redox homeostasis in a previously unknown iron-dependent manner. Using cultured differentiated C2C12 myotubes, the impact of taurine on markers of lipid peroxidation, redox-sensitive enzymes and iron-related proteins was studied. Significant increases in the heme protein myoglobin and the iron storage protein ferritin were observed in response to taurine treatment. Taurine supplementation reduced lipid peroxidation and BODIPY oxidation by ~60 and 25%, respectively. Furthermore, the mRNA levels of redox-sensitive heme oxygenase *(Hmox1), catalase (Cat) and glutamate-cysteine ligase (Gclc) and* the total cellular glutathione content were lower in taurine-supplemented cells than they were in the control cells. We suggest that taurine may inhibit the initiation and propagation of lipid peroxidation by lowering basal levels of cellular stress, perhaps through reduction of the cellular labile iron pool.

## 1. Introduction

Iron plays an important role in physiological and pathophysiological biochemical processes. The pool of iron in the body consists of heme proteins (hemoglobin, myoglobin and heme enzymes), transport and storage proteins (transferrin and ferritin) and functional iron-sulfur cluster proteins (e.g., complexes I, II and III of the respiratory chain) [1,2]. Iron-dependent proteins are important for enabling cellular growth and proliferation [3,4] and energy metabolism [5,6]. The majority of cellular iron is bound to proteins, and only a small proportion (1–2%) is free. The so-called labile iron is chelatable and includes both ferrous (Fe^2+^) and ferric (Fe^3+^) ions [7]. Labile iron acts as a catalyst for the generation of reactive oxygen species (ROS), through, for example, the Fenton reaction (Fe^2+^ + H_2_O_2_ → Fe^3+^ + OH^−^ + OH^●^), which forms highly reactive hydroxyl radical (OH^●^) species. Reactive oxygen species such as OH^●^ promote lipid peroxidation [8] and oxidative damage to other biomolecules.

Taurine is a nonproteinogenic amino sulfonic acid that is either endogenously produced from the degradation of methionine and cysteine [9,10] or is derived from dietary sources of animal origin [11]. For chemical structure see Scheme 1.

Excitatory tissues such as the retina, brain, heart and skeletal muscle [12,14,15] contain especially high concentrations of taurine. These tissues have a high metabolic rate and are highly vulnerable to oxidative damage. Moreover, aging might alter taurine homeostasis in several tissues of laboratory rodents, as shown in Table 1.

Skeletal muscle is the main storage site for taurine in the mammalian body. Interestingly, in humans [21] and other species [22,23,24,25], taurine was shown to be substantially higher in oxidative type I compared to glycolytic type II muscles. Muscle senescence and atrophy are accompanied by declining cellular taurine concentrations [20,26,27], altered iron metabolism and increased oxidative injury [28,29]. Furthermore, ageing and disease are also associated with altered iron metabolism, especially when there is an excess of labile iron. Therefore, a causal relationship between taurine and iron metabolism in muscle health and disease is plausible. Research in recent decades has revealed that taurine exhibits cytoprotective properties and prevents tissues from oxidative injury [10]. Potential molecular mechanisms of taurine, such as free-radical scavenging activity [15,30], induction of antioxidative defense mechanisms [31,32,33] and diminished generation of reactive oxygen species (ROS) [34], have been proposed. Moreover, in iron-overload rodent models, dietary taurine reduced myocardial [35] and hepatic [36,37] tissue injury caused by oxidative stress. The induction of antioxidative defense mechanisms, such as higher activity of antioxidative enzymes or elevated glutathione levels, was suggested to underlie the cytoprotective properties of taurine [35,36]. Apart from that, little is known about the potential direct effects taurine may exert on catalytic transition metals such as iron. One possibility that has been discussed by Petrova and Neudachina is that taurine may have the ability to physically bind metal ions [38]. However, experimental evidence of the physical interaction between taurine and iron has not yet been provided. Muscle cells are particularly suitable for studying the molecular effects of taurine due to the lack of key enzymes required for endogenous taurine synthesis. In the present study, differentiated C2C12 mouse myoblasts were used to examine the molecular effects of taurine on cellular iron and redox homeostasis.

## 2. Materials and Methods

### 2.1. Cell Culture

C2C12 cells were purchased from IAZ (Institute for applied cell culture, Munich, Germany). Myoblasts were maintained in Dulbecco’s modified Eagle’s medium (DMEM, high glucose 4.5 g/L, P04-03590, PAN^TM^ Biotech GmbH) supplemented with 20% (v/v) fetal bovine serum (FBS) and 1% Pen/Strep. Final experiments were conducted with differentiated myotubes. To this end, cells were seeded into cell culture plates and proliferated until they reached approximately 90% confluence (~48 h later). Myogenesis was initiated by switching the growth medium (20% FBS) to differentiation medium containing 2% horse serum. The differentiation medium was refreshed every 24 h. After four days of differentiation cells expressed the intended morphology of fused myotubes. At day four the differentiation medium was supplemented with taurine or respective other substances. If not indicated otherwise, the abbreviation TAUR means that cells were treated with 5 mmol/L taurine in differentiation medium.

### 2.2. Cell Viability Test

Cell viability was estimated via neutral red uptake assay [39]. Cells were seeded in 24-well plates and were differentiated as described in the “Cell culture” section. C2C12 myotubes were incubated with taurine (Sigma Aldrich, T0625, Darmstadt, Germany) in a range of concentrations (1–250 mmol/L) and 10% ethanol as the positive control to induce cytotoxicity. After 24 h, the cells were incubated with 1% neutral red solution (Carl Roth GmbH, Karlsruhe, Germany) for 2 h. Afterwards, cells were lysed in a discoloration solution, and adsorption of neutral red was measured with a microtiter plate reader at 540 nm (iEMS Reader MF, MTX Labsystem Inc.,Vienna USA).

### 2.3. Thiobarbituric Acid Reactive Substances (TBARS)

Lipid peroxidation was determined by colorimetric TBARS assays according to the method described by Kai (1978) [40]. Differentiated myotubes were cultured in 6-well plates and treated with 5 mmol/L taurine, 250 µM Trolox or control medium for 48 h. Fresh treatments replaced the old after 24 h. Cells were harvested by scraping with Dulbecco’s phosphate-buffer saline (DBPS) on ice. Total protein content was estimated using the bicinchoninic acid (BCA) method (Pierce^TM^ BCA Protein Assay, Thermo Scientific, Waltham, USA). Proteins of the remaining cell suspension were precipitated with 5% trichloroacetic acid (TCA). The supernatant (200 µL) was stabilized in 20 µL of 0.5% mixture of sodium docecyl sulfate (SDS)-butylated hydroxytoluene (BHT), and an equivalent volume of 1% thiobarbituric acid (TBA) was added. The mixture was incubated for 20 min at 99 °C. After cooling on ice, 1 mL of butanol was added to the mixture to extract the pink trimetin colorant. The supernatant was transferred to a 96-well plate in triplicate. Fluorescence (λex = 520 nm/λem = 560 nm) was measured with a microplate reader. Tetraethoxypropane (Sigma Aldrich, Darmstadt, Germany) was used as an external standard.

### 2.4. Lipid Peroxidation Determined by BODIPY Assay

C11-BODIPY (581/591) (Life Technologies, D-3861, Darmstadt, Germany) is a fluorescent dye that can be used as a lipid peroxidation sensor. This fatty acid analogue is lipophilic and incorporates into biomembranes. After being oxidized, the fluorescent properties of C11-BODIPY shift due to the change from the reduced (λ_ex_ = 540 nm/λ_em_ = 595 nm) to the oxidized (λ_ex_ = 480 nm/λ_em_ = 520 nm) state. The adjusted index of BODIPY oxidation was calculated as follows:Adjusted Index = Em_Ox_/(Em_Ox_ + Em_Red_)

C2C12 cells were seeded in black-walled, clear-bottom 96-well cell culture plates. C2C12 cells were first differentiated as described previously. Prior to the addition of BODIPY, cells were incubated with medium (CON) or taurine (TAUR). After 24 h, the medium was removed, and the cells were rinsed and treated with 10 µM BODIPY in medium for 30 min. Then, the medium was removed, and the cells were exposed to either 2 µmol/L iron(II) sulfate (Fe^2+^, Merck, Germany), 80 µmol/L cumene hydroperoxide (CumOOH, Sigma Aldrich, Germany) or a mixture of 2 µmol/L Fe^2+^ and 80 µmol/L CumOOH in DPBS at 37 °C for 1 h. Afterwards, cells were washed with DBPS, and fluorescence of adherent cells was measured using a microplate reader (TECAN infinite F200, Grödig, Austria).

### 2.5. Cellular Total Glutathione

Cells were treated without (CON) or with 5 mmol/L taurine (TAUR) for 24 h. After the medium was removed, adherent cells were washed with DPBS and lysed in 10 mM HCl by being subjected to freeze-thaw cycles. The total protein content in the lysates was determined by the BCA method. The remaining lysates were used for GSH detection after proteins were precipitated with an equal volume of 6.5% sulfo salicylic acid (SSA). The GSH concentration was measured according to the enzymatic recycling method described by [41,42]. In short, GSH is oxidized by 5,5’-dithio-bis(2-nitrobenzoic acid) (DTNB) to form glutathione disulphide (GSSG) and 5’-thio-2-nitrobenzoic acid (TNB). The latter can be measured at 412 nm. In the presence of nicotinamide adenine dinucleotide phosphate (NADPH), GSSG can be recycled by GSH reductase. Here, 20 µL of standards/samples were transferred to a 96-well plate and incubated for 5 min with 200 µL of reagent mix (1 mmol/L DTNB and 0.34 mmol/L NADPH in stock buffer). Then, 40 µL of GSH-reductase (8.5 IU/mL) was quickly added to each well, and adsorption was measured kinetically every 20 s for 2 min at 415 nm (iEMS Reader MF, MTX Labsystem Inc., Vienna, USA).

### 2.6. Quantitative Real Time Polymerase Chain Reaction (qRT-PCR)

Differentiated myotubes were incubated in control media in the absence of taurine (CON) or with 5 mmol/L taurine (TAUR) for 6 h. Afterwards, cells were harvested with peqGOLD TriFast (VWR International GmbH, Darmstadt, Germany), and RNA was isolated according to the manufacturer’s instructions. Sample RNA concentrations and quality were assessed with a Nano Drop 2000 UV-Vis spectrophotometer (Thermo Fisher Scientific GmbH, Life Technologies™, Darmstadt, Germany). Gene expression was analyzed via qRT-PCR with a SensiFAST^TM^ SYBR^®^ No-ROX One-Step Kit (Bioline GmbH, Luckenwalde, Germany) using a Rotorgene 6000 cycler (Corbett Life Science, Sydney, Australia). The primer sequences used for the quantification of the target genes are listed in Table 2. Relative mRNA levels were quantified as the ratio between the target gene and a housekeeping gene (*Ap3d1*).

### 2.7. Western Blotting

Cultivated C2C12 cells were differentiated and treated with CON or TAUR for 24 h. Cells were harvested on ice with a RIPA-PIC-PhosSTOP (100:1:10) buffer mixture. After cell lysis and centrifugation of the sample, the supernatants quantified via the BCA method and were then subjected to western blotting. For western blotting, 30 µg of each sample was loaded on MINI PROTEAN^®^ TGX-Stain-Free^TM^ precast gels (BioRad, Munich, Germany) and then was separated via gel electrophoresis. Afterward, the proteins were transferred to a membrane (Immuno-Blot^®^ PVDF Membrane for Protein Blotting, BioRad, Munich, Germany), blocked with skimmed milk and incubated with primary antibodies overnight. The following antibodies were used: cytochrome c, 15 kDa (sc-13560, Santa Cruz Biotechnology, Dallas, USA), ferritin heavy chain 21 kDa (sc-376594, Santa Cruz Biotechnology), ferritin light chain, 19 kDa (ab69090, Abcam, Berlin, Germany), myoglobin, 17 kDa (D2F5X, Cell Signaling, Leiden, Netherlands), taurine transporter 65-70 kDa (AB5414P, Merck, Darmstadt, Germany) and total OXPHOS antibody cocktail (ab110413, Abcam). For the taurine transporter, the target band was detected at approximately 140 kDa, indicating a dimeric structure. Afterwards, membranes were incubated with a secondary antibody for 1 hour, and bands were visualized using an enhanced chemiluminescence (ECL) substrate and a ChemiDoc XRS system (both BioRad). The signal of targeted bands was densitometrically analyzed and normalized against the total protein load. Image Lab Software 5.2.1 (BioRad, Munich, Germany) was used for protein quantification.

### 2.8. Catalase Activity

The catalase activity was determined according to Johansson and Borg (1988) [43]. The assay is based on the catalase driven oxidation of methanol in the presence of hydrogen peroxide to produce formaldehyde. Formaldehyde reacts with 4-amino-3-hydrazino-5-mercapto-1,2,4-triazole (Purpald, Sigma Aldrich, Darmstadt, Germany). After addition of alkaline potassium periodate solution, the formaldehyde-purpald adduct oxidizes and develops a color that is detected spectrophotometrically at 540 nm.

For cell culture experiments differentiated myotubes were incubated in 6-well plates with CON and TAUR. After 24 h, cells were sonicated in 200 µL of phosphate buffer and supernatant was used for the determination of catalase activity and total protein quantification via BCA method. For the present study, 30 µL samples and 66 µL assay mix (phosphate buffer, methanol, hydrogen peroxide) were incubated at room temperature for 20 min before the addition of potassium periodate. The catalase activity was calculated using a dilution series produced by purified catalase from bovine liver (Sigma Aldrich, Darmstadt, Germany).

### 2.9. Statistical Analysis

Statistical analyses were conducted using GraphPad PRISM software (San Diego, CA, USA) and IBM SPSS Statistics 24 (Ehningen, Germany). For statistical hypothesis tests, groups were analyzed for normality of the distribution (Kolmogorov-Smirnov and Shapiro-Wilk tests). In the case of normally distributed data, Levene’s test was conducted to assess the homogeneity of variances. If the null hypothesis was rejected (Levene’s test indicated it was not significant), for multiple comparison a one-way analysis of variance (ANOVA) with a one-sided Tukey test as post hoc analysis was performed. If the null hypothesis was confirmed (Levene’s test indicated significance), the Games-Howell post hoc test was used. For comparison of two groups a *t*-test was performed with normal distributed data while the Mann–Whitney U test was chosen if data did not pass the test of normal distribution.

## 3. Results

### 3.1. The Impact of Taurine on Cell Viability and Cellular Taurine Management

The cell viability of medium control (CON) cells was set as 100%. Taurine tended to be cytotoxic at concentrations ≥250 mmol/L (Figure 1a). Lower taurine concentrations (5–50 mmol/L) increased the absorbance of viable at 540 nm up to 160% compared to control. Microscopy images (Figure 1c) illustrate the morphology of already differentiated myotubes after incubation with 10% ethanol (EtOH) and different taurine concentrations. The control cells exhibited the typical morphology of long myotubes that were formed during four days myogenesis by fusion of C2C12 myoblast. The treatment with 5 and 100 mmol/L taurine did not affect cell morphology compared to control cells. The positive cytotoxicity control EtOH as well as the highest concentration of 250 mmol/L taurine seemed to reverse the differentiation status of the myotubes which is reflected by cell fragmentation. Moreover, EtOH induced the removal the adherent cells from the plate and their agglomeration within the medium. The lowest taurine concentration that enhanced the cell viability (5 mmol/L) was used for further experiments and is abbreviated hereafter as TAUR. Furthermore, the mRNA levels of the gene encoding the taurine transporter (Taut) were significantly downregulated, while the TAUT protein levels were not affected by taurine treatment (Figure 1b, Appendix A).

### 3.2. The Impact of Taurine on Celluar Redox-Homeostasis

Cellular lipid peroxidation (LPO) was determined using TBARS and BODIPY assays (Figure 2a,b). Under basal conditions, taurine significantly prevented the generation of secondary LPO products (TBARS) by approximately 60% within 48 h. For the BODIPY assay, CON-and TAUR-pretreated cells were loaded with the fluorescent dye C11 BODIPY (581/591), a fatty acid analogue that incorporates into the cell membrane. BODIPY oxidation was measured under basal conditions (without any exogenous stressors) and after 60 min of supplementation with iron (Fe^2+^) and cumene hydroperoxide (CumOOH). In general, Fe^2+^ itself did not affect BODIPY oxidation, while BODIPY oxidation was induced by CumOOH. Taurine significantly counteracted CumOOH-induced BODIPY oxidation (−25%). In addition, the mRNA levels of genes encoding redox-sensitive enzymes were quantified in control and taurine-treated cells. The mRNA levels of heme oxygenase 1 (*Hmox1*), catalase (*Cat*) and glutamate-cysteine ligase catalytic subunit (*Gclc*) were downregulated in response to taurine treatment (Figure 2c). In line with the decline of the Cat mRNA level, the catalase activity was significantly lower in taurine-treated compared to control cells. In addition, the reduced *Gclc* mRNA level detected in taurine-treated cells, was accompanied by an approximately 50% drop of the total cellular glutathione level, which consists of reduced (GSH) and oxidized (GSSG) glutathione (Figure 2d).

### 3.3. The Impact of Taurine on Iron-Related Proteins

The protein levels of four iron-related proteins were analyzed via western blotting (Figure 3a, Appendix A). The light chain ferritin isoform (FTL), which is involved in cellular iron storage, increased more than two-fold in response to taurine treatment. However, the heavy chain ferritin isoform (FTH) was not affected by taurine supplementation. The protein levels of the heme protein CYTC were slightly increased, and compared to the control, the MB protein amount was 1.7 times higher upon taurine incubation. The relative mRNA levels of *Ftl*, *Fth* and *Tfr* were analyzed, since their expression is tightly regulated by intracellular iron availability (Figure 3b). As a positive control, C2C12 cells were treated with 2 µmol/L Fe^2+^-sulfate. Iron-treated myotubes exhibited significantly higher *Ftl* mRNA expression, while *Tfr* mRNA expression was downregulated. Taurine did not affect *Ftl*, *Fth* or *Tfr* mRNA levels compared to those of the control. Furthermore, protein levels of specific subunits representing the presence of mitochondrial oxidative phosphorylation complexes were not affected by the taurine treatment (Figure 3c, Appendix A). In addition, the mRNA levels of four genes encoding proteins that are involved in iron-sulfur-cluster (ISC) protein synthesis and assembly were quantified via qRT-PCR. Taurine supplementation led to a 1.5-fold increase in mRNA levels of the *Nubpl* gene. The mRNA levels of genes encoding for *Nfu1*, *Bola3* and *Glxr5* did not differ between control and taurine treated cells (Figure 3d).

## 4. Discussion

Differentiated C2C12 mouse myotubes were used as a cell culture model to study the impact of taurine on cellular redox homeostasis and iron metabolism. Since skeletal muscle is not able to synthesize appropriate amounts of taurine, muscle cells need to take up circulating taurine via active transport to prevent deficiency symptoms such as impaired muscle function and energy metabolism [44,45]. In the present study, cultured myotubes were supplied with exogenous taurine via cell culture medium. Therefore, an appropriate, non-toxic concentration of taurine was initially established by the neutral red assay. The highest taurine concentration (250 mmol/L) significantly reduced the viability of C2C12 myotubes. Interestingly, taurine concentrations between 5 and 100 mmol/L seemed to enhance the cell viability compared to control. The neutral red assay provides information on the number of living cells which were calculated relative to the amount of living control cells. Since control cells were not exposed to external stress, we suggest that taurine did not prevent the degradation of existing myotubes but rather re-activates the proliferation of quiescent myoblasts. However, this point needs further investigation.

Physiological taurine concentrations have already been determined to be approximately 15 to 30 mmol/L in mouse skeletal muscle [46,47,48]. However, there is a substantial concentration gradient between intracellular and extracellular mammalian taurine levels in vivo [12,49]. To attain more physiological extracellular taurine levels, 5 mmol/L taurine was used for further experiments. In vivo, transcellular taurine transport occurs actively against a concentration gradient [12]. The taurine transporter (TAUT) is encoded by the solute carrier family 6 member 6 (*Slc6a6; Taut*) gene. In our study, *Taut* mRNA was significantly downregulated (by 50%) in response to supplementation of the cell culture medium with 5 mmol/L taurine. Accordingly, Han et al. reported on a concentration and time dependent downregulation of *Taut* mRNA at physiological extracellular taurine concentrations of 0.05 and 0.5 mmol/L in MDCK kidney cells [50]. Thus, taurine may generally repress the *Taut* mRNA expression via negative feedback regulation. However, the TAUT protein levels were not affected, indicating that the TAUT protein amount may not be exclusively regulated at the transcriptional level. As observed by others, the TauT activity is tightly regulated and there might be a passive efflux via volume-sensitive anion channels [51,52]. Furthermore, it cannot be excluded that taurine may act via extracellular signaling or as taurine-conjugated metabolites such as N-acyl taurine (NATs) or glutamyltaurine rather than in its free form [53].

The potential antioxidative properties of taurine in cultured C2C12 cells were determined via TBARS and BODIPY assays. Under basal conditions, long-term taurine treatment (48 h) prevented the formation of lipid peroxidation products (TBARS) by approximately 60% compared to that of the control. Furthermore, the role of taurine in the protection of membrane PUFAs against oxidation was investigated using the fluorescent fatty acid analogue C11-BODIPY (581/591) as the LPO sensor. The impact of taurine on BODIPY oxidation was tested under basal conditions or after stimulation with low doses of Fe^2+^ and CumOOH. BODIPY oxidation was not different between untreated and taurine-supplemented cells under basal conditions. BODIPY oxidation was significantly increased upon treatment with CumOOH, an agent propagating LPO in cells; this effect was significantly counteracted by taurine. Because taurine is a water-soluble molecule, it seems unlikely that it accumulates in the lipid layer of membranes to directly inhibit the propagation of LPO. On the other hand, CumOOH is a semi-stable oxidizing agent that has to be “activated” by transition metals such as Fe^2+^ to form lipophilic cumoxyl radicals (CumO^●^) [54,55]. Once it becomes a radical, CumO^●^ attacks unsaturated fatty acids or other susceptible structures (e.g., BODIPY) within the lipid layer of biomembranes [8]. Thus, taurine supplementation may diminish the intracellular labile iron pool (LIP), and consequently, reduce the conversion of the semi-stable CumOOH to the more reactive CumO^●^ radical.

In addition to the LPO biomarkers, cellular total glutathione (GSH) was reduced, and mRNA levels of genes encoding redox-sensitive antioxidative enzymes (*Hmox1, Cat* and *Gclc*) were significantly downregulated in response to taurine. Moreover, the catalase activity was reduced by approximately 40% due to the taurine treatment. These results support our hypothesis that taurine may act via suppression of oxygen radical formation (driven by free iron) rather than by upregulating defense and repair mechanisms induced by oxidatively damaged biomolecules. However, most of the assays applied in our study relate to unstressed cellular conditions. The presence of cellular stressors may also explain the contrasting results reported in the literature. Several other studies have indicated that taurine [33,56,57] and its more potent derivative taurine chloramine [31,58,59] enhance *Hmox1* expression through induction of the transcription factor nuclear factor erythroid 2-related factor 2 (Nrf2).

The impact of taurine on the protein expression of representative iron-associated proteins was determined via western blotting. Ferritin is the major iron storage protein in mammals and consists of 24 subunits and two isoforms, the light chain ferritin (FTL) and the heavy chain ferritin (FTH). FTH exerts ferroxidase activity and converts ferrous ions (Fe^2+^) into the storable ferric (Fe^3+^) form, while FTL aids in iron nucleation [60]. The arrangement of the ferritin subunit assembly is species- and tissue-specific [61]. Under the conditions investigated, taurine significantly increased the FTL protein levels more than two-fold but did not affect FTH protein levels. Interestingly, FTL-rich ferritin has been shown to accumulate more iron ions than FTH-rich ferritin [62]. Thus, the cellular storage capacity for iron may be augmented in taurine-supplemented cells, suggesting the prevention of accumulating excess labile iron. Moreover, the higher the proportion of protein-associated iron, the lower the LIP and, consequently, the extent of LPO [63,64]. Interestingly, Epsztejn et al. demonstrated that the reduction of LIP through ferritin was accompanied by a decrease in cellular glutathione content [63]. Furthermore, iron(II)glutathione was shown to be the dominant component of the labile iron pool [65]. Likewise, in our study, supplementation with taurine significantly reduced total glutathione levels in muscle cells.

Cellular LIP is sensed and regulated by iron responsive element-binding proteins (IRPs), which regulate ferritin and transferrin post-transcriptionally [66]. At low cellular labile iron levels, IRPs bind to iron-responsive elements (IREs) within the mRNA of the two ferritin isoforms and inhibit their translation. In contrast, binding of IRP to IRE of the *Tfr* mRNA inhibits its degradation and enhances transferrin translation and thus cellular iron uptake [67]. This may explain the higher *Ftl* and lower *Tfr* mRNA levels observed in our study in response to iron treatment (Figure 2B). *Ftl* and *Fth* were not different after taurine supplementation, while the protein concentration of FTL was significantly increased. It may be hypothesized that taurine supplementation modulated proteasomal FTL degradation in a regulatory mechanism that was independent of its transcript level, which has been described earlier [66,68]. Taken together, the regulation of IRP-induced modulation of LIP through taurine supplementation seems plausible and holds great potential for future studies.

Cellular iron exists beyond the free (LIP) in stored (ferritin) forms, which is crucial for the synthesis of iron-dependent proteins such as heme or iron-sulfur (Fe-S) cluster proteins. In our study, the levels of the heme proteins myoglobin and cytochrome c were significantly higher after taurine supplementation. This may further emphasize the shifting of cellular iron from labile forms to stored forms and protein association in response to taurine supplementation. In vivo, the majority of iron is bound to the heme protein hemoglobin in red blood cells, and there is evidence for a regulative role for taurine in this process. Oral iron treatment for iron-deficiency and anemia in women was more effective when taurine was co-supplemented [69], and circulating taurine was positively correlated with the amount of red blood cells in female endurance athletes [70].

Other mitochondrial proteins that require iron for assembly, including the Fe-S cluster proteins of the respiratory chain, were not affected by taurine. The mRNA level of *Nupbl* was increased, although it was the only gene affected among the tested genes that encoded mitochondrial Fe-S cluster proteins. To date, taurine has been suggested to affect the protein synthesis of the mitochondrially encoded subunits mt-ND5 and mt-ND6 [34]. It remains unclear whether taurine may modulate Fe-S protein cluster synthesis systematically and thereby modulate mitochondrial function.

## 5. Conclusions

We provide evidence that taurine treatment of cells, which rely on exogenous taurine supply, reduces the extent of LPO, which we assume is caused by shifting of cellular iron from free to protein-associated forms. The involvement of labile iron pool sensing proteins and the role of taurine in mitochondrial Fe-S cluster protein synthesis and respiratory function warrant further investigation. It would be also interesting to validate the effect of taurine in cells and tissues with high iron stores and/or substantial iron metabolism, such as hepatocytes or osteoblasts. Moreover, tissue taurine levels, including skeletal muscle, are often diminished in aged versus young control animals [16,17,18,19,20]. Therefore, future studies should address the questions, whether dietary taurine supplementation affects redox- and iron-homeostasis during the aging process.

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
