# Peer review of "Taurine Enhances Iron-Related Proteins and Reduces Lipid Peroxidation in Differentiated C2C12 Myotubes"

_antioxidants, 2020, doi:10.3390/antiox9111071_

Round 1

Reviewer 1 Report

The topic of the manuscript “Taurine Enhances Iron-Related Proteins And Reduces Lipide Peroxidation In Differentiated C2C12 Myotubes” falls with the scope of Antioxidants. This work provides that the taurine treatment of C2C12 cells reduce the extent of LPO and could be associated by shifting of cellular iron from free to protein-associated forms.

We recommend to the authors to check the following aspects:

Abstract page 2

Page 3 line 79 “foetal bovine serum” replaced by fetal bovine serum”

            line 98 “DBPS” replaced by Dulbeco’s phosphate-buffer saline (DBPS)”

           line 101 “BHT” replaced by butylated hydroxytoluene (BHT)”

Page 5 Table 2  The nucleotide primers used for taurine transporter are missing

           lines 150 “Fth” replaced by“Ftl

Page 8 line 240 “taurine did not affect Ftl, Fth or Tft mRNA levels”

           Is Tfl mRNA level slightly higher?

           line 244  “ISC protein” replaced by “ Iron sulfur cluster (ISC) protein”

           It is mandatory to check all above mentioned issues.

Therefore, I recommend paper acceptance after minor revisions.

Author Response

The topic of the manuscript “Taurine Enhances Iron-Related Proteins And Reduces Lipide Peroxidation In Differentiated C2C12 Myotubes” falls with the scope of Antioxidants. This work provides that the taurine treatment of C2C12 cells reduce the extent of LPO and could be associated by shifting of cellular iron from free to protein-associated forms.

We are grateful to the very favorable comments of the reviewer.

We recommend to the authors to check the following aspects:

  • Page 3

line 79 “foetal bovine serum” replaced by “fetal bovine serum”

line 98 “DBPS” replaced by “Dulbeco’s phosphate-buffer saline (DBPS)”

line 101 “BHT” replaced by “butylated hydroxytoluene (BHT)”

We have now changed these three text passages as suggested by the reviewer.

  • Page 5 Table 2

The nucleotide primers used for taurine transporter are missing

The primer pair for Taut is now included in table 2 of the revised version as suggested by the reviewer.

  • Page 8

line 240 “taurine did not affect Ftl, Fth or Tft mRNA levels”. Is Tfl mRNA level slightly higher?

The impact of taurine on Ftl and Tfr mRNA levels was not statistically significant.

line 244  “ISC protein” replaced by “ Iron sulfur cluster (ISC) protein”

Done.

Reviewer 2 Report

The manuscript studies the role of taurine in enhancing iron-related proteins and reducing lipid peroxidation in differentiated c2c12 myotubes. The manuscript was well written and easy to understand. However, I have a few comments:

  1. Please describe the use of differentiated cells in more details. What are the phenotypes of the differentiated cells? Do they undergo oxidative stress, apoptosis, cellular damage, etc? How soon after switching to 2% horse serum do the differentiated cells exhibit the pathologies? Considering you have selected 5mM taurine as it increases the cell viability, have you looked at apoptosis and oxidative stress in C2C12 myotube cells before and after switching to horse serum? In other words, what is the basis of using differentiated cells in this study?
  2. The authors measured the mRNA levels of redox-sensitive heme oxygenase, catalase and glutathione-cysteine ligase as well the total glutathione content. Have you measured the protein levels or activities of these (or any other) antioxidant defenses? mRNA and protein levels may increase in response to taurine treatment, but there may be no change in antioxidant activities. It will be interesting to see if taurine treatment increase the antioxidant activities in C2C12 myotube cells.
  3. Please correct: Table 1: Long Evan rats brain(stratium). I think you meant to say “striatum.”
  4. Figure 1c: Please describe the morphological changes in the results section.
  5. Lines 182-183: Taurine tended to be cytotoxic at concentrations…”
  6. Lines 275-265: Taurine concentrations between 5 and 100 mmol/L did not adversely affect the viability of C2C12 cells. Based on Figure 1a, these concentrations, except 100mmol, significantly increase the viability when compared to control. Please clarify.
  7. In addition to the cell viability effect, have you looked at other parameters when determining the ideal physiological concentration of taurine? Have you looked at the mRNA levels of TauT in cells treated with various doses of taurine?

Author Response

The manuscript studies the role of taurine in enhancing iron-related proteins and reducing lipid peroxidation in differentiated c2c12 myotubes. The manuscript was well written and easy to understand. However, I have a few comments:

We are grateful to the very favorable comments of the reviewer.

  • Please describe the use of differentiated cells in more details.

We have extended the cell culture differentiation protocol in the material and method section of the revised version as suggested by the reviewer.

What are the phenotypes of the differentiated cells? Do they undergo oxidative stress, apoptosis, cellular damage, etc? How soon after switching to 2% horse serum do the differentiated cells exhibit the pathologies?

Please see below representative pictures of C2C12 cells during differentiation from day 0 (start of differentiation/ i.e. two days after seeding) until day 4 (start of treatments with taurine etc.). At day 4 of differentiation, C2C12 cells typically exhibited the intended phenotype (fused myotubes):

The cells did not show any pathologies during differentiation. As shown in Figure 1c of the manuscript, cells only exhibited pathologies at day 5 after treating the already differentiated cells for 24 h with EtOH and 250 mmol/L taurine (cytotoxic concentration).

Considering you have selected 5mM taurine as it increases the cell viability, have you looked at apoptosis and oxidative stress in C2C12 myotube cells before and after switching to horse serum? In other words, what is the basis of using differentiated cells in this study?

We did not analyse markers of apoptosis and oxidative stress in undifferentiated cells. We applied a more muscle fiber like cell culture model instead of proliferating myoblasts since it has been reported that muscular taurine depletion diminishes muscle health and exercise performance (Ito et al., J Amino Acids. 2014; 2014:964680. doi: 10.1155/2014/964680). Furthermore, Uozumi et al. (2006) observed an increased TauT mRNA expression during myogenesis indicating increasing demand of taurine in differentiating or differentiated cells (Biochem J, 2006 Mar 15;394(Pt 3):699-706. doi: 10.1042/BJ20051303).

  • The authors measured the mRNA levels of redox-sensitive heme oxygenase, catalase and glutathione-cysteine ligase as well the total glutathione content. Have you measured the protein levels or activities of these (or any other) antioxidant defenses? mRNA and protein levels may increase in response to taurine treatment, but there may be no change in antioxidant activities. It will be interesting to see if taurine treatment increase the antioxidant activities in C2C12 myotube cells.

This is an important point raised by the reviewer. In fact, mRNA and protein as well as enzyme activity levels may not be regulated in the same direction. We included mRNA levels of Gclc, Hmox-1 and Cat because they are direct targets of the transcription factor Nrf2, which is a key regulator of cellular redox-homeostasis. In good accordance with an Nrf2-mediated effect of taurine supplementation, these three Nrf2 target genes showed a similar decline in the mRNA expression level. Furthermore, the reduction of both, the glutamate-cysteine ligase (Gclc) mRNA levels and corresponding cellular glutathione levels provided consistent results.

The protein expression of HMOX1-1 was not affected by the taurine treatment (as indicated below). Thus, taurine-mediated effects on Hmox-1 mRNA level were not reflected on the protein level. Posttranslational regulatory mechanisms such as a long protein half-life may be responsible for this discrepancy.

Measuring the enzyme activity of redox-sensitive enzymes, such as catalase is very important to gain more insights into the role of taurine in the regulation of redox-homeostasis. We now included results of the catalase activity assay to the manuscript as suggested by the reviewer (Figure 3d).

  • Please correct: Table 1: Long Evan rats brain (stratium). I think you meant to say “striatum.”

Corrected.

  • Figure 1c: Please describe the morphological changes in the results section.

We now extended the description of morphological changes in the result section as suggested by the reviewer.

  • Lines 182-183: Taurine tended to be cytotoxic at concentrations…”

We now included “at”

  • Lines 275-265: Taurine concentrations between 5 and 100 mmol/L did not adversely affect the viability of C2C12 cells. Based on Figure 1a, these concentrations, except 100mmol, significantly increase the viability when compared to control. Please clarify.

We conducted the neutral red assay to evaluate a toxic taurine concentration. The enhancement of the cell viability due to taurine was rather a secondary finding which may be difficult to interpret. An explanation for this observations could be that taurine rather re-activated the proliferation of non-differentiated quiescent myoblasts than enhanced the viability of existing differentiated myotubes. However, this point needs further investigation.  We now added a comment in the respective paragraph of the discussion section of the revised version of the manuscript.

  • In addition to the cell viability effect, have you looked at other parameters when determining the ideal physiological concentration of taurine? Have you looked at the mRNA levels of TauT in cells treated with various doses of taurine?

This is an important point raised by the reviewer. We have looked at TauT mRNA levels after 6 h incubation with decreasing taurine supplementation from 100->50->25->5 mmol/L as summarized in the figure below.

Irrespective of the taurine concentration used, the TauT mRNA levels were decreased by approximately 50 % compared to control cells. Unfortunately, we did not use concentrations lower than 5 mmol/L taurine. Han et al (1997) observed a downregulation of Taut mRNA levels in MDCK cells after 24 h treatment with already 0.05 mmol/L taurine (~20 % decrease) and 0.5 mmol/L (~50 % decrease) taurine (Biochimica et Biophysica Acta 1351 (3): 296–304; DOI: 10.1016/S0167-4781(96)00217-5). Thus, we assume that high physiological (≥ 0.5 mmol/L) or supra-physiological (≥1 mmol/L) circulating taurine concentrations generally downregulate Taut mRNA levels in skeletal muscle via a negative feedback mechanism.

Reviewer 3 Report

In this manuscript the authors test the hypothesis that taurine can inhibit the initiation and progression of lipid peroxidation through direct interaction of taurine with the intracellular iron pool.

The topic is interesting and original. The experiments are very well designed and appropriate to answer the research question, which is clearly defined.  The results are statistically evaluated and clearly understandable.  The authors could show that taurine modulates the portion of labile iron to stored protein associated iron and thus diminishes lipid peroxidation.

The paper is very well written and I don´t see any inconsistencies in the interpretation of data.  Taken together the scientific aspect addressed in this study is novel and the paper is well suited to be published in Antioxidants.

Minor comment:

The authors should provide a mechanistic explanation why even low concentrations of exogenous taurine significantly improve cell viability based on the results that taurine mRNA was significantly downregulated while taurine protein levels remain unaffected.

It would be interesting to see whether the result are consistent using other cell line e.g. osteoblasts. The authors should briefly comment on a “universal” validity of their findings.

Author Response

Comments and Suggestions for Authors

In this manuscript the authors test the hypothesis that taurine can inhibit the initiation and progression of lipid peroxidation through direct interaction of taurine with the intracellular iron pool.

The topic is interesting and original. The experiments are very well designed and appropriate to answer the research question, which is clearly defined.  The results are statistically evaluated and clearly understandable.  The authors could show that taurine modulates the portion of labile iron to stored protein associated iron and thus diminishes lipid peroxidation.

The paper is very well written and I don´t see any inconsistencies in the interpretation of data.  Taken together the scientific aspect addressed in this study is novel and the paper is well suited to be published in Antioxidants.

We are grateful to the very favorable comments of the reviewer.

Minor comment:

  • The authors should provide a mechanistic explanation why even low concentrations of exogenous taurine significantly improve cell viability based on the results that taurine mRNA was significantly downregulated while taurine protein levels remain unaffected.

Physiological plasma taurine concentrations in mice are approximately in the range of 0.1‑1 mmol/L. The concentration of 5 mmol/L taurine applied to the cultured cells as in the present study was relatively high which may have led to a negative feedback regulation of Taut mRNA expression. Han et al (1997) observed a downregulation of Taut mRNA levels in MDCK cells after 24 h treatment with already 0.05 mmol/L taurine (~20 % decrease) and 0.5 mmol/L (~50 % decrease) taurine (Biochimica et Biophysica Acta 1351 (3): 296–304; DOI: 10.1016/S0167-4781(96)00217-5). Thus, we assume that high physiological (≥ 0.5 mmol/L) or supra-physiological (≥1 mmol/L) circulating taurine concentrations generally downregulate Taut mRNA levels in skeletal muscle via negative feedback. Nevertheless, there should be a sufficient TAUT activity leading to cellular taurine accumulation, which causes the observed effects.

We now added a short text passage regarding the regulation of cellular influx and efflux of taurine.

  • It would be interesting to see whether the result are consistent using other cell line e.g. osteoblasts. The authors should briefly comment on a “universal” validity of their findings.

This is an important point raised by the reviewer. We added a comment in the conclusion addressing the above mentioned point regarding the “universal” validity of our findings.

Round 2

Reviewer 2 Report

The authors have addressed my concerns.